# Changes in Sacrifice by Burning and the Transfer of the Space Inhabited by Ghosts in China: Philological and Linguistic Perspectives

**Cong Li** [1] and **Yiyun Zhang** [2,*]

1　Department of Chinese Language and Literature, Peking University, Beijing 100871, China; bnulicong@163.com

2　School of Chinese Language and Literature, Beijing Normal University, Beijing 100875, China

*　Correspondence: yyzhangchina@126.com

**Abstract:** This paper analyzes changes in sacrifice by burning and the space inhabited by ghosts in ancient China from philological and linguistic perspectives. During the Shang and Zhou dynasties, rulers believed that they could convey their offerings and reverence to their ancestors in heaven by burning firewood and sacrifices (燎). From the Spring and Autumn Period to the Han dynasty, the ancient Chinese metaphors for naming the underground space inhabited by ghosts experienced a transformation from a natural space (Yellow Spring (黄泉)) to human settlements (li (里), big cities (都)) and then to government institutions for criminal penalty (government (府), prisons (狱)), which symbolized the gradual establishment of a living order in the space inhabited by ghosts based on the human society. When the new living order of the space inhabited by ghosts was established, the ancient Chinese began to reconstruct sacrifice by burning during the Wei and Jin dynasties, and the objects burnt were represented by joss paper. The use of the term "transforming (化)" to refer to sacrifice by burning suggests that people believed that burning with fire was a way to transfer objects from the real world to the world of ghosts. The act of burning joss paper not only embodied the Chinese concept of ancestor worship to "treat the dead as if they were alive" but also gave "fire (火)" rich religious connotations while greatly simplifying the process and cost of sacrificial rituals, thus gradually becoming popular.

**Keywords:** sacrifice by burning; ancient China; space of ghosts; Chinese character; metaphor

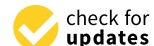

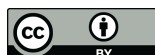

## 1. Introduction

Ancient Chinese people believed that sacrifice by burning offered a means of communication between people and ghosts. The sacrificial ritual of burning objects with special symbolic meanings dates back to the Shang dynasty. Of course, religious rituals of burning offerings exist in communities across the world, but an interesting element of sacrifice by burning in China is that the objects burnt and the symbolism of burning have changed radically over the course of history. At the very beginning, the Chinese burnt objects during sacrificial rituals to express their reverence to their ancestors and deities and pray for blessings from them. As recorded in the *Book of Documents·The Canon of Shun* (《尚书·舜典》), "Shun offered burnt sacrifices to the six Honoured ones. (禋于六宗。)". Furthermore, the *Discourses of States·Discourses of Zhou I* (《国语·周语上》) elucidates that such sacrificial acts were characterized by the expression, "Offer with purity and reverence. (精意以享，禋也。)". During rituals, firewood was burnt, sometimes along with jade, silk, and sacrifices; then, later on, many of the same objects as those in the world of the living appeared among the objects burnt during rituals, such as household utensils, coins, and clothing, as well as symbolic objects made of paper, most typically joss paper, which were burnt for deceased families to use in the underworld.

One paradox is that people in the Shang and Zhou dynasties offered sacrifices by burning because they believed that the smoke produced by burning would carry their reverence upwards to deities and ancestors in heaven, but why was the ritual of burning retained and held even more widely when it was believed that deceased families resided in the underworld? What's more interesting is that to denote the act of burning offerings with fire, this means of sacrifice was called "burning (燎)" during the Shang and Zhou dynasties, while it acquired the new term "transforming (化)" later on, which is still used by the Chinese in the naming of containers known as "money transforming basins (化钱盆)" or "treasure transforming basins(化宝盆)" for burning offerings during sacrificial rituals today. The ideological concepts represented by the words "burning (燎)" and "transforming (化)" are also utterly different. A study from the linguistic and philological points of view should help us to further understand the transformation of the concept of religious sacrifice in China.

Much attention has been paid to the Chinese imagination of the afterworld and ghosts, especially the Taoist and Buddhist conceptualization of the netherworld. However, as a means of communication between the real world and the afterworld, sacrifice by burning has not received sufficient attention, especially in terms of diachronic changes. As a form of sacrifice that has been used from ancient times to the present, sacrifice by burning has persisted through the development history of religion in China, and its changing and enriched forms and connotations can be taken to represent changes in the spiritual world of the Chinese.

Focusing on sacrifice by burning, this paper studies its formal and conceptual changes in China, especially through the lens of philology and linguistics, and analyzes changes in the Chinese imagination of the space of ghosts and relevant social concepts behind this imagination.

## 2. Communication with Divinities: The Original Meaning of Sacrifice by Burning

"To treat the dead as if they were alive (事死者如事生)" was an important ancient Chinese notion on how the dead should be treated, and burnt sacrifices were part of it. "To treat(事)" means to serve someone properly, and in the view of ancient people, to treat the dead as if they were alive was a sign of filial piety and respect for them.

According to archaeological data, there were a large number of tools for life and production, ritual utensils, carriages, horses, people, etc., buried as funeral objects in tombs of specific forms in the Shang and Zhou dynasties (Ou 1986; Xu and He 2004). This seems to indicate that ancient people in a very early period believed that the dead would need to use these people or objects just as they did when they were alive. Then, with the development of the ritual system, this simple religious concept was given humanistic and moral connotations:

*The Book of Rites· The Meaning of Sacrifices*, "In sacrifices, King Wen served the dead as if they were alive. He missed the dead as if he did not wish to live. On their death days, he was sad; when hearing their names, he looked as if he saw them." (《礼记·祭义》："文王之祭也：事死者如事生，思死者如不欲生，忌日必哀，称讳如见亲。")

*The Book of Rites· The Doctrine of Mean*, "Each one stands in his place, performs the ritual of sacrifice, plays the music of sacrifice, honors the ancestor he should honor, loves the ancestors he should love, and serves the departed ancestors as if they were alive. This is the highest standard for filial piety. "(《礼记·中庸》："践其位，行其礼，奏其乐，敬其所尊，爱其所亲，事死如事生，事亡如事存，孝之至也。")

*Xunzi·On Rites*, "The rite of funerals is to decorate the dead and see them off as if they were alive. Thus, we should always follow the rite whether to treat the dead or the living". (《荀子·礼论》："丧礼者，以生者饰死者也，大象其生以送其死也。故事死如生，事亡如存，终始一也。")

In the order of the ceremonial system, the religious rite of arranging an elaborate funeral for the dead to show respect and remembrance became a requirement at a ritual and moral level.

In addition to tombs, people in the Shang and Zhou dynasties also established fictional spaces inhabited by the spirits of the deceased, and rulers believed that the deceased kings continued to exist in heaven, with sacrifice by burning a means of communication between the earthly and the heavenly. From records in oracle bone inscriptions, it can be seen that the ritual of "burning (尞)" (later written as "燎") already appeared in the Shang dynasty. There are two oracle bone glyphs for "burning (尞)": one is "𣏟" (*Collection of Research on Bone Shell Inscriptions 1200*, abbreviated hereinafter as the *Collection*), with "wood (木)" in the center, on both sides of which the dots symbolize sparks during burning and indicate the burning of firewood; the second glyph is "𤓠" (*Collection* 19712), which adds the image of "fire (火)" under the first glyph, to further highlight the act of burning with fire. In oracle bone inscriptions, "burning (尞)" is a ritual of sacrifice by burning in memory of late kings, gods of nature, and deities of all kinds, including some deceased officials. Below are some examples.

Six heads of cattle were burnt as sacrifices to Nao. (尞于夒六牛。) (*Collection* 14369)

Through divination, ten heads of cattle were burnt as sacrifices to Shangjia by the river. (贞，尞于上甲于河十牛。) (*Collection* 1186)

Through divination, a pig was burnt as a sacrifice to Huangyi. (贞，尞黄尹豕。) (*Collection* 3477)

Through divination on the day of yi-hai, a sacrifice by burning was held to the Earth God, and it rained. (乙亥卜，尞于土，雨。) (*Collection* 22048)

From the cattle, a head was burnt as a sacrifice to the God of the West. (尞于西一牛。) (*Collection* 14329)

The glyphs of the Chinese character show that "wood" is burnt in the ritual of "burning (尞)", during which sacrifices such as "cattle (牛)" and "pigs (豕)" used to be offered, according to oracle bone inscriptions.

Besides "burning (尞)", people of the Shang dynasty also had a ritual of burning people with fire, known as "burning humans (烄)". The oracle bone character for "burning humans (烄)" is written as "𤆍" (*Collection* 30789) or "𤐫" (*Collection* 15674), which consist of two parts: a person with legs crossed and a fire underneath. The ritual "burning humans (烄)" could be used to offer sacrifices to ancestors, though more often, it was used as a way of praying for rain:

With divination on the day of ding-wei, the sacrifice by burning was held to worship Mugeng, and then it [rained]. (丁未卜，烄囗母庚，又从[雨]。) (*Bone Shell Inscriptions Tunnan* 3586)

With divination on the day of yi-hai, the sacrifice by burning was held to worship Zuding. (乙亥卜，其于祖丁，其烄。) (*Collection* 27306)

Through divination, the sacrifice by burning was held, and then it rained. (贞烄，有从雨。) (*Collection* 15674)

The sacrifice of burning was held today, so it rained. (今日烄，有雨。) (*Collection* 29993)

It was a rooted concept to burn people for rain in the Shang dynasty, and the Duke Xiang of Song, a descendant of Shang, still wanted to burn "witches" in order to seek rain hundreds of years after the fall of the Shang dynasty. As recorded in *Zuo Zhuan·The 21st Year of Duke Xiang*, "During a severe drought in the summer, the Duke wanted to burn witches. (夏大旱，公欲焚巫尪。)" However, he was eventually stopped by his minister Zang Wenzhong.

The rulers of the Shang dynasty were keen on sacrifice by burning. In addition to "burning (燎)" and "burning humans (烄)", there were also rituals such as "firewood (柴), binding (束), fetching (取), and chopping (新)" (Wang 1982). At the Shang Dynasty Ruins in Zhengzhou, a pit and a surface of burnt earth were excavated (Henan Provincial Institute of Cultural Heritage and Archaeology 2001, p. 496), which many researchers believe were sites used by the rulers of the Shang dynasty for sacrifice by burning (Pei 1991; Xie 2019). When we explore the cognition of the Shang rulers, this reveals their belief that the late kings resided in heaven and sacrifice by burning could transport what they wanted to ask, along with the offerings, to heaven; then, in turn, it would bring them instruction and protection from heaven and their ancestors.

The reason for adopting sacrifice by burning as a means of communication between "heaven (天)" and the real world was related to the understanding of "fire (火)" in the Shang dynasty. The most typical glyph for "fire (火)" in oracle bone inscriptions is "🔥" (*Collection* 9104), which depicts the shape of a flame traveling upward. Of course, if this is only an objective facsimile of a physical phenomenon, while "hot (炎)" and "emperor (帝)" are more expressive characters for the upward movement of "fire (火)". First, "hot (炎)" is written as "🔥" in oracle bone inscriptions (*Collection* 36512), which is the shape of two "fires (火)" stacked vertically. According to *Shuowen Jiezi·Section of Yan* (《说文解字·炎部》), "the character 'hot (炎)' refers to the upward light of fire. (炎，火光上也。)". In the *Book of Documents·Hong Fan* (《尚书·洪范》), "Fire is known as upward heat.(火曰炎上。)". As Kong Yingda quoted from Wang Su, "The nature of fire is hot, bright and upward." This reflects how the ancient Chinese believed that an important characteristic of fire was its upward movement.

The character "emperor (帝)" appeared in different forms such as "🔼" (*Collection* 14307) and "🔽" (*Collection* 30388) in oracle bone inscriptions, in which "emperor (帝)" was used as a sacrificial ritual:

With divination on the day of wei, a sacrificial ritual was held for Xiayi. (未卜，帝下乙。) (*Collection* 22088)

With divination on the day of gui-chou, a sacrificial ritual was held in the east. With divination on the day of gui-chou, a sacrificial ritual was held in the south. (癸丑卜，帝东。癸丑卜，帝南) (*Collection* 34145)

Wang and Bai (2016) proposed that the character "Di (帝)" meant gathering firewood for sacrifice by burning, i.e., piling up firewood and burning that to offer sacrifices to heaven, which was a ritual later written as "Di (禘)". The difference between "Di (帝)" and "burning (燎)" is that on top of the former, there is one more horizontal stroke than for the latter in oracle bone inscriptions. A horizontal stroke is also written on top of the character "heaven (天)" to symbolize the uppermost part of the character (*Collection* 36557). According to Wang and Bai (2016), when considering "heaven (天)" as a referential symbol, we can see that "Di (帝)" resided in heaven and had the characteristic of being high above in the minds of the ancient Chinese. We can surmise that it was because both "heaven (天)" and "Di (帝)" symbolized supremacy that the rulers of the late Shang dynasty began to attach the character "Di (帝)" in front of the names of their ancestors, such as "Di Yi (帝乙)" and "Di Xin (帝辛)", and "Di (帝)" transformed from a sacrificial ritual in the beginning into the names of Chinese monarchs later on.

When analyzing the glyphs of oracle bone inscriptions and the meanings of the words involved, it can be seen that people believed heaven and their ancestors were up above, and "fire (火)" was a form of upward mobility in the Shang dynasty, so they had a particular preference for sacrifice by burning. An interesting case is the story where "Zhou (纣)", the last king of the Shang dynasty, cremated himself, as recorded in the *Records of the Grand Historian—Basic Annals of the Shang Dynasty* (《史记·殷本纪》), "King Wu of Zhou then attempted to lead the vassals to attack King Zhou. King Zhou also sent his

troops to Muye. On the day of Jia-zi, when his army was defeated, King Zhou went back to the Lutai Palace, put on his precious jade garment, and burnt himself to death. (周武王于是遂率诸侯伐纣。纣亦发兵距之牧野。甲子日，纣兵败。纣走，入登鹿台，衣其宝玉衣，赴火而死。)" According to Chang (2017), the account of the "precious jade garment (宝玉衣)" suggests that King Zhou might have completed some sacrificial rite that people of the Shang dynasty believed in before his death. It can be said that sacrifice by burning ran throughout the Shang dynasty, and even the king who caused the fall of his dynasty wished to burn himself to apologize to heaven and his ancestors. His status as king made him a supreme human sacrifice in the sacrificial ritual, and enacting the ritual of "burning humans (燎)" upon himself also served as self-punishment for causing the dynasty to fall.

The Zhou dynasty continued with the traditional sacrifice by burning. *Yi Zhou Shu·Shi Fu* (《逸周书·世俘》) recorded King Wu holding a sacrifice by burning to worship his ancestors after conquering the Shang dynasty: "King Wu presided over the sacrifice, and the prime minister carried the white flag with the head of King Zhou of Shang and the red flag with his wife's head into the Zhou Temple, and burnt their left ears as sacrifices there". More often than not, sacrifice by burning in the Zhou dynasty was called "Yan (禋), Chai (柴), Jiao (槱)".

> In the *Book of Documents·Luo Gao*, it is remarked, "I dare not enjoy it, but offer(禋) it to King Wen and King Wu". (《尚书·洛诰》：予不敢宿，则禋于文王武王。)

> In the *Book of Rites·System of Rule*, the following description is given: "In the second month, the king should begin a tour eastward, reach Mount Tai first, and burn firewood(柴) and offerings to worship famous mountains and rivers". (《礼记·王制》：岁二月，东巡守，至于岱宗。柴而望，祀山川。)

> In the *Book of Rites·Da Zhuan*, the following event is recorded: "The victory of King Wu over King Zhou of Shang at Muye was a major event in his establishment of the Zhou Dynasty. After the war, a sacrifice by burning(柴) was held to the gods of heaven and earth, and sacrifices were offered to ancestors in the dormitories at Muye. (《礼记·大传》：牧之野，武王之大事也。既事而退，柴于上帝，祈于社，设奠于牧室。)

> In the *Rites of Zhou·Dazongbo*, it is stated that people could "Worship Great Heaven by the sacrifice by burning incense (禋祀), the celestial deities by burning sacrifices (实柴), and the gods of Sizhong, Siming, Wind and Rain by burning firewood (实柴)". (《周礼·大宗伯》：以禋祀祀昊天上帝，以实柴祀日、月、星、辰，以槱燎祀司中、司命、飌师、雨师。)

Zheng Xuan's note reads, "The three sacrificial rituals all piled up firewood and filled with sacrifices, or jade and silk, and burnt them to create smoke". Zheng Xuan suggested the three sacrificial rituals were similar in "piling up firewood and filling with sacrifices", which was the same as sacrifice by burning in the Shang dynasty. It seems people of the Zhou dynasty also believed that the power of fire could convey their reverence upward. However, there were two changes in sacrifice by burning in the Zhou dynasty compared with that in the Shang dynasty. On the one hand, the people of the Zhou dynasty no longer held sacrifice by burning as frequently as those of the Shang dynasty, but only on important occasions (Kang 2023); on the other hand, the people of the Zhou dynasty were not as keen on burning living people and witches to seek rain as before, which was why the Duke Xiang of Song was stopped when trying to burn witches to seek rain, as mentioned above.

In summary, during the Shang and Zhou dynasties, there was a conceptualization that ancestors occupied a celestial realm. Rulers of the period initiated the construction of sanctified sites dedicated to the spirits of the deceased, colloquially termed "heaven (天)". Ritualistic sacrifices were performed by means of combustion to venerate both the ancestors and deities of this heaven. The role of "fire (火)" was believed to serve as a conduit, elevating the offerings and reverence up to heaven for their ancestors and the gods to enjoy. However, the imagination of where ancestors lived in this period was chaotic; it

was only known that, like the gods, they received sacrifices in heaven, while how they lived in the space for spirits was not clearly imagined.

It must be noted that in the "heaven (天)" constructed by the rulers of the Shang and Zhou dynasties, not everyone's departed families could reside and accept sacrifices. In both the Shang and Zhou dynasties, only the king's ancestors and a very few important officials could partake in sacrifice by burning, which means that the space for the spirits of the dead was accessible only to a tiny minority of people. Neither should it be overlooked that the fuel of burning was firewood in sacrifice by burning during the Shang and Zhou dynasties, and when an offering had to be thrown into the fire, it had to be "physical (实)", that is, a real object used by the living, quite different from the burning of symbolic objects later on.

### 3. From the Yellow Spring (黄泉) to an Underground Government (土府) and Hell (地狱): Shifts in the Space of Ghosts Embodied by Metaphorical Naming

Although the rulers of the Shang and Zhou dynasties constructed a heavenly space for spirits through sacrifice by burning, the space was a privilege for the kings of Shang and emperors. During the Spring and Autumn Period, with the devolution of political power, dukes, princes, and ministers also began to imagine the space for ghosts after death. They chose the underground as the space for this form of life after death, and two of the representative events were the interpretation of the "Yellow Spring (黄泉)" by Duke Zhuang of Zheng and the discussion about "Nine Plains (九原)" between Zhao Wenzi and Shu Xiang, two ministers of Jin.

> In *Zuozhuan·Yingong 1st Year*, it is said, "Thus, he arranged for his mother Jiang to live in Chengying, and vowed never to see her until he reached Yellow Spring! Then, he regretted it." (《左传·隐公元年》：遂寘姜氏于城颍，而誓之曰：不及黄泉，无相见也！既而悔之。)

> In the *Book of Rites· Tangong II*, Zhao Wenzi and Shu Xiang visited Nine Plains. Wenzi said, "If one can get up and work again after death, who would be the most virtuous that We can go back to cooperate with?" (《礼记·檀弓下》：赵文子与叔向观乎九原。文子曰："死者如可作也，吾谁与归？") (The same event is also recorded in *Discourses of States· Discourses of Jin VIII* (《国语·晋语八》))

Duke Zhuang of Zheng arranged for his mother to live in "Chengying (城颍)" because she favored his younger brother, and said, "I will not meet my mother again until I reach Yellow Spring". The narration reflects a preconception: people at that time believed that they could enter the space known as "Yellow Spring (黄泉)" after death, where they could meet their loved ones. Zhao Wenzi and Shu Xiang were both ministers of the State of Jin, so they were colleagues. When they visited "Nine Plains (九原)", a cemetery in the State of Jin where late ministers were buried, they discussed the people with whom they would work after death. In this conversation, three words are worth noting. The first is "if (如)", a hypothetical conjunction, which means that the class of ministers then did not yet fully believe that there was a space for their own ghosts to reside in after death; the second is "get up and work(作)", which means getting up and moving around, suggesting that they assumed they could get up to carry out activities after death as if they were alive underground; the third is "return (归)", which was used to give a name to the life after death despite its original meaning of "going back (回去)", suggesting that people at the time had already begun to think of death as the continuation of life, and begun to regard "life (生)" and "death (死)" as a cycle.

During the Spring and Autumn Period, the aristocracy postulated that a subterranean domain was the abode for their spirits post mortem. This was because, on the one hand, the heavenly space was already occupied by rulers of a higher status, and on the other hand, the underground was a place for burying corpses. This paradigm shift allowed even those of relatively lower classes of people to envisage a distinct spiritual existence beyond death. The emergence of this concept signified a departure from the earlier Shang dynasty be-

lief, which predominantly envisioned ancestors as celestial entities who merely received earthly offerings. The new belief system imbued spirits with the agency to engage in activities within their designated spectral sphere, thus expanding the scope of afterlife existence beyond mere passive reception of sacrifices.

In the Warring States Period, the "Yellow Spring (黄泉)" was more widely used to refer to the afterworld:

> As recorded in *Strategies of the Warring States—Strategies of Chu I* (《战国策·楚策一》), it is recounted that "Anling Jun sobbed several times and said, 'I sit next to you in the palace, and go out in the same carriage with you. When you pass away, I would like to be your mat in Yellow Spring, so that the mole crickets and ants won't come to disturb you, and nothing could be happier than that!' The King was so pleased that he formally bestowed the title Anling Jun on him". (安陵君泣数行而进曰："臣入则编席， 出则陪乘。 大王万岁千秋之后，愿得以身试黄泉，蓐蝼蚁， 又何如得此乐而乐之。" 王大说，乃封坛为安陵君。)

> In *Guanzi·Xiao Kuang*(《管子·小匡》), it is recorded that, "bestowed by the Duke, even though I was dead in Yellow Spring, my soul would be immortal. (应公之赐，杀之黄泉，死且不朽。)"

The "Yellow Spring (黄泉)" might have been the lowest space that people in the Spring and Autumn Period and the Warring States Period could imagine. "Yellow (黄)" is the color of the earth, and "Yellow Spring (黄泉)" originally referred to underground springs. As recorded in *Zhuangzi·Tian Zifang*, "All superb masters can demonstrate their skills at ease whether they are in the blue sky or in Yellow Spring, calm and composed. Your nervous, frightened and panic eyes reveal how fearful you are now. (夫至人者，上闚青天，下潜黄泉，挥斥八极，神气不变。今汝怵然有恂目之志，尔于中也殆矣夫！)" Here, the terms "Yellow Spring (黄泉)" and "blue sky (青天)" are used in contrast to indicate the bottom and the top of the space in which the "superb masters" carry out activities. This understanding is the result of ancient people observing and summarizing the basic elements of the earth. In the *Book of Rites·Doctrine of Mean* (《礼记·中庸》), it is written that:

> "The way of Heaven and Earth may be summed up in a word. They are undistracted, so produce unfathomable things. The way of Heaven and Earth is broad and profound, superb and brilliant, far-reaching and long-standing. The Heaven now before us is only a bright stretch, but the sun, the moon and stars outspread and all things are covered in its infiniteness exist. The earth before us is just a handful of soil, but it is so broad and thick that it carries mountains like Mount Hua without feeling their weight and contains the rivers and seas without leaking. The mountain now before us appears only a stone, but it is so vast that grass and trees grow on it, birds and beasts inhabit it, and precious treasures are found in it. The water now before us seems a ladleful; yet in its unfathomable depth live the largest tortoises, alligators, dragons, fishes, and turtles, and abound goods and wealth. (天地之道，可壹言而尽也。其为物不贰，则其生物不测。天地之道，博也厚也，高也明也，悠也久也。今夫天，斯昭昭之多，及其无穷也，日月星辰系焉，万物覆焉。今夫地，一撮土之多，及其广厚，载华岳而不重， 振河海而不泄，万物载焉。今夫山，一拳石之多，及其广大，草木生之， 禽兽居之，宝藏兴焉。今夫水，一勺之多，及其不测，鼋鼍、蛟龙、鱼鳖生焉，货财殖焉。)"

This passage states basic elements from heaven to earth that were in the minds of ancient people, namely "heaven (天)", "earth (地) (soil) (土)", "mountain (山) (stone) (石)" and "water (水)" in a top-down sequence. The ancient Chinese called the underground water a "spring (泉)", so the lowest position in space was named the "Yellow Spring (黄泉)". The use of the term "Yellow Spring (黄泉)" to refer to the space for the dead was then the result of using the lowest space in cognition as a metaphor for the afterworld.

During the Warring States Period and the Han dynasty, some people called the space where the dead resided "Gloomy City (幽都)". In *Elegies of the South·Requiem*

(《楚辞·招魂》), Qu Yuan exclaimed, "Oh, soul, come back! Don't go to the Gloomy City down below. (魂兮归来！君无下此幽都些。)". Here, he is calling the ghosts of the dead back from "Gloomy City (幽都)", which he describes as "down below (下)", indicating that the people of Chu also believed that the space for ghosts after death was underground. The original meaning of "gloomy (幽)" is deep and dark, and the periphery of the character "gloomy (幽)" is in the shape of a "mountain (山)", thus referring to the space deep underground in *Elegies of the South*. In ancient Chinese, "city (都)" could refer to a large and important town or a unit for the control of the population, and the appearance of the name "Gloomy City (幽都)" suggests that people at that time already believed that the ghosts of the dead would gather in the underground world. The back wall of the Xiaotangshan Stone Shrine (Han dynasty) was found to have patterns in the underground space surrounded by diagonal lines, diamond shapes, and copper coins. According to Jiang (2016, pp. 78–86), the Han paintings depict the world after death, and the basic nature of the burial chamber is a space in "Gloomy City (幽都)", with the patterns in the mural paintings symbolizing "soil (土)", "stone (石)", and "spring (泉)", which represent a cross-section of the earth and reveal the underworld below the "Nine Springs (九泉)", as shown in Figure 1.

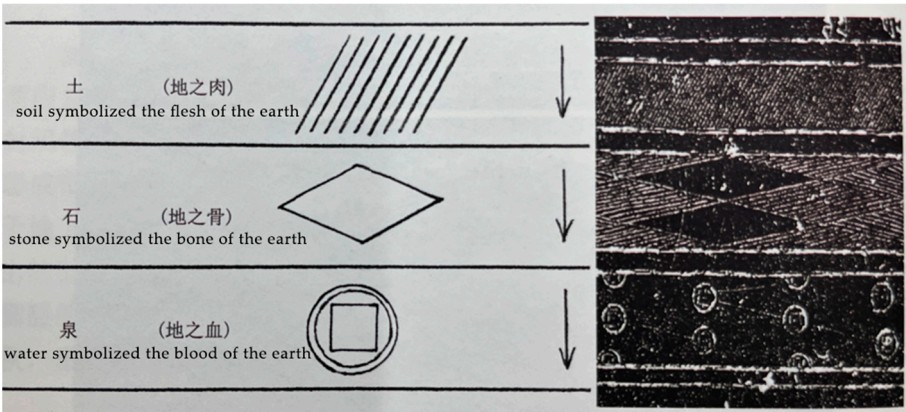

**Figure 1.** Patterns in the Xiaotangshan Stone Shrine and their meanings (Jiang 2016, p. 86).

Originally, a "spring (泉)" referred to water coming naturally from underground, which was used as a symbol on money and copper coins because the ancient people believed that money and the spring shared the same character: circulation. As recorded in the *Rites of Zhou·Offices of Earth·Spring Government* (《周礼·地官·泉府》): "Spring Government is in charge of levying taxes and collecting the unsold goods and the goods that are overstocked in civil use". (泉府掌以市之征布、敛市之不售、货之滞于民用者。) In the *Rites of Zhou·Offices of Heaven·External Government* (《周礼·天官·外府》), there is "the charge of the national circulation of money". Zheng Xuan of the Han dynasty noted, "Money is just like the spring. Its pronunciation is contained in that of announcement in Chinese. Its treasury is called the spring, and its circulation is known as distribution. It is named after the spring, and can be circulated everywhere. (布，泉也。布，读为宣布之布，其藏曰泉，其行曰布。取名于水泉，其流行无不遍。)". In this spirit, the cognition of connections between the "spring (泉)" and currency became prominent in the Han dynasty, and Wuzhu coins were commonly used to symbolize the "spring (泉)" in portraits at the time (Jiang 2016, pp. 81–85).

Because the "Yellow Spring (黄泉)" was considered to be the lowest position, with the gradual formation of the doctrine of Yin and Yang, it came to be regarded as the place with the strongest "Yin Qi (阴气)", as well as the place where the Yang Qi began to sprout because of the transformation between Yin and Yang. Here are some examples:

> In *Yi Zhou Shu·Zhou Yue Jie*, it is written, "On the day of the Winter Solstice in the 1st month, the constellation Mao and Bi can be seen in the sky at dusk; the day is the shortest, and then gradually becomes longer, with a slight movement of Yang

Qi under the ground, the Yin Qi on the ground condensing, and everything withering away." (《逸周书·周月解》：惟一月，既南至，昏，昴毕见，日短极，基践长，微阳动于黄泉，阴降惨于万物。)

In *Huainanzi·Tianwen Xun*, the notion is recorded that "When Yin Qi peaks, it can be felt at the North Pole and down at Yellow Spring, so it is not time to drill wells". (《淮南子·天文训》：阴气极，则北至北极，下至黄泉，故不可以凿地穿井。)

In *Baihutong·Rites and Music*, the time of the year was as follows: "Xun is in the 11th month, and means fuming and steaming, with Yang Qi emerging from beneath Yellow Spring." (《白虎通·礼乐》：埙在十一月，埙之为言勋也，阳气于黄泉之下，勋蒸而萌。)

Today, the Chinese still refer to the underworld after death as the "Yin Space (阴间)". An exploration of the term "Yellow Spring (黄泉)" shows that the naming of the world after death as "Yin Space (阴间)" is not a metaphor for the direct correspondence between "Yin and Yang (阴阳)" and "up and down (上下)", but rather the metonymy of the "Yellow Spring (黄泉)" with "Yin (阴)" because of the strongest qi of Yin at the "Yellow Spring (黄泉)". As the concept of Yin and Yang became popular, the "Yin Space (阴间)" was gradually used more frequently to refer to the underground world of ghosts.

The other two names for the afterworld in the Han dynasty were "Hao Li (tomb) (薧里)" (later written as "蒿里") and "Xia Li (下里)". In the *Book of Han·The Biographies of Emperor Wu's Five Sons* (《汉书·武五子传》), Liu Xu's song of death reads: "Hao Li is summoning me, and I am doomed to death with my body exposed at the gate of the city." (蒿里召兮郭门阅，死不得取代庸，身自逝。). As recorded in the *Book of Han·The Biographies of Harsh Officials*, "Jiao and Jia rich in Maoling spent a huge sum of money secretly accumulating and storing charcoal, reeds and other materials for burial in Xia Li." (茂陵富人焦氏、贾氏以数千万阴积贮炭苇诸下里物。). Furthermore, Meng Kang (State of Wei) noted, "The dead were buried in tombs underground, which was thus called Xia Li". (死者归蒿里，葬地下，故曰下里。). As defined in Xu Shen's *Shuowen Jiezi* (《说文解字》), "Hao (薧), Li of the Dead (死人里). (Hao means a gathering place for the dead.)" The character "hao (tomb) (薧)" is composed of "grass (草)" on the top and "death (死)" at the bottom, indicating that it is the underground space for the dead. "Hao Li (tomb) (薧里)" is an imaginary name for the afterworld based on the characteristic of tombs covered by overgrown grass (Jia 2014). We are particularly interested in "Li (里)" in the interpretation, which in ancient Chinese was used to refer to a human settlement and a unit for the administration of the population. As recorded in the *Rites of Zhou·Offices of Earth·Suiren*, "Five families are a neighborhood, and five neighborhoods are Li". (五家为邻，五邻为里。). The expressions "Hao Li (蒿里)", "Xia Li (下里)", and "Li of the Dead (死人里)" are metaphors for the place where the dead gather based on the administrative units of the living. This means that the people of the Han dynasty already had the idea that the dead gathered in a public space as if they were alive, and that the world after death also followed a certain order of administration.

When the "government (府)" was used to name the residence of ghosts, this marked a further establishment of order in the space inhabited by ghosts during the Han dynasty. According to Yu (2005, pp. 92, 144), the "Heavenly Government (天府)" and the "Underground Government (土府)" used in the *Scriptures of Great Peace* (《太平经》) were projections of the government in the mortal world, and the department responsible for specific management was called "Cao (曹)", which was borrowed directly from the government agency of the Han dynasty. Around the 1st century A.D., Mount Tai started to be regarded as the place of death, and the administrator of the underworld was called the "Governor of Mount Tai (泰山府君)". Yu (2005, pp. 148) proposed that the term "Governor (府君)" should not be taken as "Lord (主)" in a general sense, but instead as a common name for county-level officials in the Han dynasty. As to why Mount Tai was called the home of the dead, we assume in this paper that it was made the supreme symbol of "earth (地)" due to the Fengshan Sacrifice rites conducted by ancient emperors of China on Mount Tai in the

Qin and Han dynasties. As the world of ghosts constructed since the Spring and Autumn Period was underground, with the gradually established order of life in the underworld in the Han dynasty, especially the prevailing idea that ghosts needed to be governed, there were more and more metaphors and connections between the world of ghosts and the real world. Therefore, as a symbol of earth in the real world, Mount Tai became the place where ghosts were in charge. This assumption is in line with the basic idea of the dead being underground, which was held since the Spring and Autumn Period.

After the introduction of Buddhism, "Hell (literally "underground prison" in Chinese) (地狱)" began to be used to name the world of ghosts. The substitution of "Hell (地狱)" for the "Underground Government (土府)" and the "Government of Mount Tai (泰山府)" basically completed the Chinese imagination of the underworld, and even after the introduction of Christianity, the term "Hell (地狱)" was still retained. Compared to the "Netherworld (地府)", the Buddhist image of "Hell (地狱)" is a link in the cycle of karma, which reflects profound changes that has occurred in the traditional concepts of the soul, life, and death since the Wei and Jin dynasties (Wei 2007).

The "prison (狱)" in ancient Chinese means litigation, cases, imprisonment, and jail, all of which are events or things related to illegal, judicial, and law enforcement behaviors. The word "Hell (地狱)" is also a metaphorical name for the space of people who find themselves within such institutions on Earth.

So far, we can see that from the Spring and Autumn Period to the Eastern Han dynasty, the space inhabited by ghosts was always named using on metaphors from the world of the living. Accordingly, the underworld was named "spring (泉)", "gloom (幽)", "li (里)", "underground (土)", and "earth (地)", which are all related to the land.

Admittedly, from a linguistic point of view, as the most basic means of cognition, it is not surprising that metaphors were used to name things, but if we observe "Yellow Spring (黄泉)", "Gloomy City (幽都)", "Hao Li(蒿里)", "Underground Government (土府)", "Netherworld (地府)", and "Hell (地狱)" from a diachronic perspective, we may find two changes in the ancient Chinese understanding of the space of ghosts:

First, a metaphor of a natural direction was initially established to denote the space of ghosts with the "Yellow Spring (黄泉)", the lowest end of nature; later, social institutions of the human world were used to metaphorically refer to the space of ghosts, such as the units of settlement and administration "city (都)" and "li (里)", and the administrative agencies of "government (府)" and "prison (狱)".

Second, in the space of ghosts established with metaphors, the order of administration gradually became stricter. In the early days, the "Yellow Spring (黄泉)" only marked the location, rather indicated the order of life; when the terms "Gloomy City (幽都)" and "Hao Li (蒿里)" appeared, this meant that ghosts gathered in a certain range, which was a simulation of human settlements; the emergence of the "Underground Government (土府)" and the "Netherworld (地府)" then suggested that the space of ghosts also needed to be administrated like the space of the living, and the term "Hell (地狱，literally an "underground prison" in Chinese)" indicated that the punitive institutions of human society had been introduced to the space of ghosts.

Combined with the order of late kings in heaven established by the Shang and Zhou rulers, changes in the metaphors for the space of ghosts can be taken to reflect the gradual enrichment of classes in the space of ghosts, and especially its accessibility to lower classes: The supreme rulers in heaven and the general nobility underground first formed stratification, and the naming of ghost administrators in the underworld implied that there was already a "food chain" to the classes in the space of ghosts. The opening of the underground space provided the general public with a habitat for their ghosts after death, refining the imagination of the afterworld with the real world as a source.

**4. Burning Joss Paper and "Transforming(化)": The Meaning Transfer of Sacrifice by Burning and the Simplification of Rituals**

From the Shang and Zhou dynasties to the Eastern Han, Wei, and Jin dynasties, the space of ghosts constructed by the ancient Chinese shifted and expanded from heaven to the underworld, with more and more lower-class groups owning their own residential space for ghosts after death. Moreover, with the real world as a metaphor, the order of life in the space inhabited by ghosts was gradually established. As the residential space of ghosts shifted and expanded, the means of passing objects from the real world to departed families in the residential space of ghosts became an issue that needed to be reconsidered during sacrifices. As this transformation occurred, the methods and significance of sacrificial burning also changed.

One way to pass items to the space of ghosts was to offer burial objects. In early Chinese tombs, objects used by the deceased during their lifetime, which were mainly pottery household utensils, were often buried along with them. By the Shang and Zhou dynasties, non-utilitarian utensils symbolizing rites and status, such as the vessels ding (鼎) and jue (爵), appeared as burial objects (Lu 1996). From the Spring and Autumn Period and the Warring States Period, Chinese burial objects gradually became more luxurious and reached a climax during the Qin and Han dynasties (Lu 1996). In addition to larger quantities and more ornate decoration of objects, metal coins also became indispensable as burial objects. There were two underlying reasons. First, the awareness of "rites (礼)" and "filial piety (孝)" was gradually strengthened, and "treating the dead as if they were alive (事死如事生)" transformed from simple worship into a code of conduct; second, with the continuous construction of order in the space of ghosts, people came to believe that they would need more goods to sustain their lives in the netherworld.

Although the custom of burying coins in tombs already existed before the Qin dynasty (Zhao 1991), it became highly popular and even a bit exaggerated to bury large quantities of metal coins in tombs during the Han dynasty. In the tomb of the Marquis of Haihun, a large number of metal coins was found, and from a money vault in the north cloister of the tomb, 285 pieces of cake-shaped gold, 33 pieces of horseshoe-shaped gold, 20 gold plates, 15 pieces of kylin-toe-shaped gold, 378 pieces of gold utensils, and over 10 tons of Wuzhu coins were unearthed (Liu 2017; Wang and Li 2019). In the Han dynasty, the buried metal coins were even given a proper name: "burial money (瘗钱)". Furthermore, as recorded in Sima Qian's *Records of the Grand Historian·The Biographies of the Harsh Officials* (《史记酷吏列传》), during the reign of Emperor Wu of the Han dynasty, "there were thieves who stole burial money in Xiaowenyuan. (会人有盗发孝文园瘗钱。)"

After the Eastern Han dynasty, the custom of elaborate funerals was somewhat less popular. Although precious metal coins were still buried directly in tombs, they were no longer as plentiful as before; instead, people began to use joss paper in place of real currency, and the custom of burning joss paper emerged from the Northern and Southern dynasties and became popular during the Tang and Song dynasties (Wang 1998; Zhao and Yang 2005; Chen and Luo 2006; Wang and Li 2019). When considering the transition from burying real money to burning paper-based substitutes for money, Wang (1998) proposed that the invention of paper in the Eastern Han dynasty was the premise and the reason for the emerging act of burning joss paper. In addition, the religious concept of Buddhism was another reason, as the Buddhist concept of the void of vanities downplayed the custom of burying real objects in tombs; instead, the burning ritual was regarded as sacred and able to transmit things from the real world to the underworld. Furthermore, Yu (2005, pp. 99–100) pointed out that the spread of grave robbing during the Han dynasty could offer evidence of opposition to the custom of elaborate funerals. We can speculate that the substitution of paper-based symbolic money in place of real money was not only influenced by religious concepts but also a means to avoid theft in the social context of rampant grave robbing.

In any case, hundreds of years later, a new sacrificial ritual of burning emerged in China, that is, burning joss paper. Compared with the Shang and Zhou dynasties, the sacrificial ritual of burning retained the use of fire as a means of communication between the

living and ghosts, but changed in two ways: First, the objects burnt changed from firewood and sacrifices to symbolic paper objects. Second, the space with which "fire (火)" communicated also changed, as people hoped to transfer the objects on earth to heaven by means of fire and smoke at the very beginning, while later, they sought to pass currency from the real world to the netherworld by means of fire. Wang Jian, a poet in the Tang dynasty, wrote in his poem Walk at Cold Food Festival (《寒食行》), "With no fire ignited for three days, how can joss paper reach Yellow Spring?" (三日无火烧纸钱，纸钱那得到黄泉。). This reveals that the burning ritual was already considered a necessary act to pass money to the netherworld.

As mentioned in Section 2, people in the Shang and Zhou dynasties believed that they could communicate with their ancestors through fire because fire and smoke float upwards. Then, why could "fire" be a bridge between reality and the space inhabited by ghosts when spirits had already moved underground? The sacredness of "fire" in Buddhism may be an influential factor (Wang 1998), but followers of Taoism also have the ritual of burning joss paper. We may find an answer to this question in the name for the act of burning paper in ancient times.

In Chinese, the act of burning involving life, death, ghosts, and gods came to be known as "transforming (化)". In the Chinese vocabulary, the words "fire (火)" and "transforming (化)" are closely related, as they have the same phonetic consonants. Moreover, they were considered to be semantically related in the Han dynasty. As recorded in *Shiming* (literally "Interpretation of Names"), "Fire is the same as transforming." (火，化也。). Likewise, according to *Baihutong* (a compilation of debates on Chinese classics), "Fire refers to transforming". (火之为言化也。). The original meaning of the word "transforming (化)" is change, and "fire (火)" can change the form of objects, so it is also called "transforming (化)". To this day, the change of something reduced to ashes is still described as "transforming into ashes (化为灰烬)" literally in Chinese. In Yin and Yang and the Five Elements, there is the idea that "fire produces earth (火生土)", precisely because the ashes left behind after burning are categorized as a kind of "earth (土)". Since the Qin and Han dynasties, "earth (土)" has been regarded as where ghosts reside, so that the product of "fire (火)" can enter the netherworld under "earth (土)", which is consistent in the doctrine of Yin and Yang and the Five Elements.

In ancient Chinese, all the words "move (动)", "vary (变)", and "transform (化)" can summarize the act to "change (变化)", but "transforming (化)" often represents a fundamental change in the form and nature of something. As recorded in the *Book of Rites·The Doctrine of Mean* (《礼记·中庸》): "If people are moved, they will give up an evil way of life and return to the path of virtue; in this way, they will be transformed into virtuous people". (动则变，变则化，唯天下圣诚为能化。). Thus, "transforming (化)" can be used to refer to the death or the birth of a person, the transformation of something from one form to another, or the entry of a person into another space:

Transforming—death. As written in *Mengzi·Gongsun Chou II* (《孟子·公孙丑下》), "And for the dead, prevent the earth from getting near to the bodies. (且比化者，无使土亲肤。)". Zhu Xi noted, "Transforming means death. (化者，死者也。)".

Transforming—pregnancy. As discussed in Master Lyu's *Spring and Autumn Annals·Indecorum* (《吕氏春秋·过理》), "(King Zhou) dissected a pregnant woman to see her transforming(fetus); killed Bi Gan to see his heart. ((纣)剖孕妇而观其化；杀比干而视其心。)". Gao You noted, "Transforming refers to pregnancy. (化，育也。)".

Transforming—Changing form. *Zhuangzi·Carefree Wandering* (《庄子·逍遥游》) described the following: "In the northern ocean there is a fish, called Kun, the length of which is immeasurable. When Kun changes into a bird, it is called Peng. (北冥有鱼，其名为鲲。鲲之大，不知其几千里也。化而为鸟，其名为鹏。)".

Transforming—Entering the immortal world. As written in *Elegies of the South·The Far-off Journey* (《楚辞·远游》), "They become immortal and disappear, but their

names will be widely known and passed down eternally. (与化去而不见兮，名声着而日延。)".

Transforming—Transforming between reality and virtuality and between different things. As discussed in *Zhuangzi·On Levelling All Things* (《庄子·齐物论》), "I do not know whether I was then a man dreaming I was a butterfly, or whether I am a butterfly dreaming I am a man. Between a man and a butterfly, there must be a difference. This is called the transformation of things. (不知周之梦为蝴蝶与，蝴蝶之梦为周与？周与蝴蝶，则必有分矣。此之谓物化。)".

Traditionally in Taoism, the act of "incinerating (焚烧)" has a special meaning of conveying messages to the netherworld. The Taoist scriptures *Taishang Lingbao Jingming Feixian Duren Jingfa·Volume 4·Chapter Renji* (《太上灵宝净明飞仙度人经法·卷四·仁济章》) (said to be from the Western Jin) record the process of burning talismans in Taoism:

> "Order deities of earth and guards to see the soul off, take it out of the netherworld, protect it from the five sufferings and the eight difficulties, let the seven ancestors ascend to immortality, keep it away from ghost officials forever, and ferry it to Zhuling, where it is tempered for a new life". The Master said, The stamp can rescue trapped ghosts. If there is a ghost, burn the talisman for commanding ghosts first, and then burn this talisman. Upon releasing the soul, passing the trials of all heavens should be written down. Whenever releasing the soul of a disciple, trials of all heavens should be passed at the same time, and write talismans in ink on yellow paper. ("勅制地祇，侍衞送迎，拔出地户，五苦八难，七祖升仙，永离鬼官，魂度朱陵，　受鍊更生。"　师曰：　此章可以救沉滞。　凡有幽魂，则先烧遣鬼呪符，次焚符。追度之时，其文当写如历关诸天关子叙事意。凡为弟子追度，则同历关诸天同为之，以黄纸墨写符文。)

This is an early and specific record of burning talismans to cast spells, which was already commonly mentioned in the same scriptures. The talisman mentioned here could save a soul that had been stranded on the way to the netherworld for a long time by sending deities of earth through a talisman to command ghosts, and then it allowed them to take the soul away from the netherworld governed by ghost officials through the recitation of incantations in conjunction with the talisman. It is worth noting here that fire was used as a procedure for communicating with the netherworld, which is similar to the later practice of burning joss paper.

The Taoist classic version of *Taishangdong Xuanlingbao Jiuku Miaojing* (*Scripture for Salvation from Distress*, 《太上洞玄灵宝救苦妙经》) (Northern Song) recorded a story of Mo Daozu burning a memorial to the throne to the netherworld in 1124:

> In Jingde Zen Temple in Chizhou, a little monk named Mo Daozu missed his mother who passed away, and recited scriptures daily in deep gratitude for his mother…He asked through Gongcao, and vowed to recite the Scripture for Salvation from Distress and the name of Heavenly Lord devoutly day and night and 1000 times, respectively, and then incinerate the scriptures. At night, his mother told him in a dream, "You've fulfilled your filial duties by reciting the scriptures and the name of Heavenly Lord devoutly, and I've gone to heaven under the talismanic order of the Supreme God. From now on, I'll never see you again, so you should make efforts independently and keep the true faith in mind". Afterwards, she flew away on a cloud. (池州城下景德禅院有僧童莫道祖，偶缘母亲倾逝，追慕不已，日诵本教经荐擢，用酬罔极之恩。…道祖即告功曹，求受《太上救苦经》及天尊号，日夜虔诚，兼而诵之，各积千徧，　具疏焚化。是夜，母亲见梦于道祖曰："汝今不负鞠育之恩，诵经持号，诚格高眞，蒙上帝符命，我已得生天矣。自后无复见汝，汝宜自勉力，勿忘正教。"言讫，乘云冉冉而去。)

Broadly speaking, the story is about a little monk named Mo Daozu who missed his mother who had passed away with an untimely death. It tells that he recited Buddhist sutras to help her ascend to heaven. However, when he asked the Gongcao, who could

save the ghosts of the dead, if she has made it to heaven, he found that his mother was still in custody in the underworld. Mo Daozu and the Gongcao then asked for clarification and recited the *Scripture for Salvation from Distress* (《救苦经》)—as well as the name of the Heavenly Lord Taiyi 1000 times, according to his mother's instruction. Then, the Gongcao reported to the underworld, and his mother was able to ascend to heaven, as she told Mo Daozu in a dream. To achieve that happy ending, in the process of communicating with the underworld, the talismanic order and a memorial to the throne were sent to the underworld by means of burning. In particular, the term "incinerate (焚化)" was used when the memorial to the throne was delivered, which shows that incineration was a relatively fixed means of transformation in the Taoist context.

Of course, cremation is popular in Buddhism, which advocates letting fire burn around to bless deceased monks and followers. In ancient times, as the term "transforming (化)" was used to refer to the act of transferring information from the world of the living to the underworld, burning during cremation was also called "transforming (化)". For example, Zhao Shuxiang of the Song dynasty wrote in *Kenqing Notes·Burning Bones to Ashes* (《肯綮录·火骨成灰》), "After the cremation, all the bones were reduced to ashes, and could not be picked up. (化讫，收其骨殖皆成灰，不可拾。)". Moreover, the act of burning joss paper could also be referred to separately as "transforming (化)":

> In Chapter 13 of *Journey to the West*, it is written that, "After burning portraits of gods and Buddhist documents for releasing ghosts, the Buddhist rite was finished, so all went to bed". (《西游记》第十三回："化了众神纸马，烧了荐亡文疏，佛事已毕，又各安寝。")

> In Chapter 32 of *Water Margin*, "There is a sedan chair on the road, followed by several people, who are carrying two cases, to burn joss paper at the cemetery". (《水浒传》第三二回："大路上有一乘轿子，七八个人跟着，挑着两个盒子，去坟头化纸。")

Clearly, "transforming (化)" could already denote birth and death, the change of form and nature, and transmission between real and virtual spaces in the pre-Qin period. When the doctrine of Yin and Yang and the Five Elements began to prevail, fire was endowed with the capability of "transforming (化)", so burning became a way of transmitting objects and information to the space inhabited by ghosts. At that time, sacrifice by burning focused on the characteristic of the act of "transforming (化)" in that it was able to travel to unreal space. Later on, "transforming (化)" was directly defined as the act of transmitting objects and information to the unreal world.

With the burning of monetized joss paper, "paper money" was used in the netherworld earlier than in the world of the living in China, where Chinese paper money did not come into use until the Song dynasty. This may seem counter-intuitive, but it precisely reflects the realistic need for a non-material currency and the sacrificial need for a substance that could be burnt among the masses, as the space inhabited by ghosts continued to expand and the right to offer sacrifices kept shifting downwards. The act of burning joss paper significantly reduced the cost of sacrifice, simplified the process of sacrifice, as well as had the logically self-consistent historical connotation of transferring objects to the space inhabited by ghosts, so it became increasingly popular.

However, after the act of burning joss paper to be transmitted to the netherworld was developed, it did not mean that the Chinese forgot the gods residing in heaven; at this time, "burning incense (焚香)" filled the gap in conveying reverence to the heavenly space. During the Spring and Autumn Period and the Warring States Period, fragrance was a symbol of purity. In Qu Yuan's *The Songs of Chu* (《楚辞》), it is written that "I wear sweet grass on shoulders, and weave and hang orchids on my waist". Later, people created fragrance by lighting spices. Yu Xin of the Northern Zhou dynasty wrote in *Horseback Archery in the Hualin Garden on the 3rd Day of the 3rd Month* (《三月三日华林园马射赋》), "Wine is carried in the carriage of attendants, and incense is burnt on double paths. (属车酾酒，复道焚香。)". As the system of ghosts and gods was gradually constructed,

the smoke produced by burning incense became a symbol of the immortal world, and the function of its burning was deemed similar to that of sacrifice by burning. As recorded in *Yi Lin* at the turn of the Western Han dynasty and the Eastern Han dynasty, "Qin lost a great territory, as the River God blamed them for dropping a piece of jade when crossing the river before; due to the River God's wrath and refusal to bless, just like woven brocade without patterns, burning incense failed to create fragrance". This was the first time in Chinese literature that "burning incense" was explicitly recorded as a sacrifice to the gods. During the Northern and Southern dynasties, the incense-burning ritual was adopted in Taoist activities such as praying for deceased ancestors on the 1st and 15th days of each lunar month, allowing the sacrificial ritual of burning incense for ancestors on these two days each month to take shape. Buddhist incense burning and the tradition of sacrifice on the 1st and 15th days of each lunar month since the pre-Qin period were combined to eventually form the incense-burning culture in China (Long 2017). Furthermore, it is worth noting that, like the burning of joss paper, the development of the incense-burning culture was also a manifestation of the lower classes gaining the right to offer sacrifices, as the cost of "incense (香)", also a means of transmitting reverence to heaven, was much lower than that of sacrifices required in the ritual of burning (燎).

In light of the ancient Chinese conception of the soul's dwelling space, it can be observed that the ancients in China gradually internalized the livings' ethical considerations for the deceased. They used their experience of the living world to imagine the afterlife, endowing it with a dimensionality and variability akin to the real world, yet it always preserved the core principle of "treating the dead as if they were alive. (事死如事生)". With the establishment of a new order of the soul, new methods of transmitting objects to the spiritual realm were also established. Furthermore, as times have evolved, the contemporary Chinese practice of burning paper replicas of modern commodities such as bank-issued currency, iPhones, laptops, and cars during sacrificial rites may initially appear comical. However, these practices are, in fact, a continuation of the ancient Chinese ritualistic concept that we should "treat the dead as if they were alive. (事死如事生)". This act of "transforming (化)" represents the contemporary development of transferring materials to the non-physical realm, a practice that has also gained popularity in other parts of the world.

## 5. Conclusions

From the initial "burning (燎)" to the later "transforming (化)", understanding has developed among the Chinese of sacrifice by burning: when the late kings resided in heaven, burning played a role in offering sacrifices upwards, and with the subsequent imagination of the space for the dead, burning then became a way of sending objects from the real world to the unreal world.

The changes in sacrifice by burning reflect differences in the construction by the Chinese of the space inhabited by ghosts. At the very beginning, the Chinese believed that the spirits of their deceased ancestors resided in heaven. As the Chinese then went on to conceive of the space inhabited by ghosts, it gradually came to be believed that departed families had the same life in the underground world as the living did. The naming of the underground space inhabited by ghosts by the ancient Chinese took on the use of metaphors based on the cognition of the real world, and the sources of such metaphors changed from natural spaces to human institutions, i.e., from the "Yellow Spring (黄泉)" symbolizing the lowest physical space to "li (里)" and "big cities (都)" symbolizing units of human settlements, and then to "government agencies (府)" and "prisons (狱)" symbolizing administrative bodies, which served to gradually establish order in the underground space inhabited by ghosts.

As belief grew in the space inhabited by ghosts, people endowed sacrifice by burning with new forms and meanings based on fire. It can be said that the act of burning joss paper after the Wei and Jin dynasties combined the pre-Qin tradition of burning sacrifices, the Confucian requirement of filial piety and rite of treating the dead as if they were alive, the Taoist understanding of fire in the Five Elements, and the Buddhist sanctification of fire.

Moreover, owing to its low cost and the transfer of the right to sacrifice to common people, burning joss paper has become the most common and characteristic form of sacrifice in China.

**Author Contributions:** Conceptualization, C.L. and Y.Z.; methodology, C.L.; validation, C.L. and Y.Z.; formal analysis, C.L.; investigation, C.L. and Y.Z.; resources, C.L. and Y.Z.; writing—original draft preparation, C.L.; writing—review and editing, Y.Z.; visualization, Y.Z.; supervision, C.L.; project administration, C.L.; funding acquisition, C.L. All authors have read and agreed to the published version of the manuscript.

**Funding:** This research received no external funding.

**Institutional Review Board Statement:** Not applicable.

**Informed Consent Statement:** Not applicable.

**Data Availability Statement:** Data are contained within the article.

**Conflicts of Interest:** The authors declare no conflict of interest.

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
