# Peer review of "Changes in Sacrifice by Burning and the Transfer of the Space Inhabited by Ghosts in China: Philological and Linguistic Perspectives"

_religions, doi:10.3390/rel15020158_

Round 1
Reviewer 1 Report
Comments and Suggestions for Authors
This is a very fine piece. My work in Daoism has focused on Daoist ritual, including the burning of ritual objects, as well as the traditions about Haoli Shan below Mt. Tai, so I found the essay both interesting and very well argued.
Comments on the Quality of English LanguageThere are only a few rough patches and sentences that I noticed and I don't want these to detract from the outstanding article. Some of these are in the abstract. Perhaps these are examples: line 6 say "analyzes" not "analyzed". Line 11, "big cities" not "big city". Line 12, make it "prison" "prisons". Line 20, make "rich enough religious connotations" say simply "rich religious connotations" During the main text, most composition issues are not distractive, but line 237 should be rephrased. Does line 235 "bread and thick" need to be "broad and thick"?
Author Response
Thank you for your comment. Please see the attachment.

Reviewer 2 Report
Comments and Suggestions for Authors
The paper was interesting and should be published. I recommend getting a good English speaking editor to tighten the prose.
Two (very) small things stood out. The author says several times that "sincerity" was being offered in the Shang and Zhou sacrifices, but nowhere is this substantiated by a quote or reference. Second, several references are made to things like "in Buddhist concepts." "In Buddhism" will do. "Concepts" is superfluous.
Comments on the Quality of English LanguageAgain, there are one or two places where the prose was not at all clear. I would not want these minor language issues to distract the reader's attention from what is otherwise a very interesting article.
Author Response

(The authors gave the same response as above.)
